# Different soluble aggregates of Aβ42 can give rise to cellular toxicity through different mechanisms

Suman De[1], David C. Wirthensohn[1], Patrick Flagmeier [1,2], Craig Hughes[1,3], Francesco A. Aprile[1,2], Francesco S. Ruggeri[1,2], Daniel R. Whiten [1], Derya Emin[1], Zengjie Xia[1], Juan A. Varela [1], Pietro Sormanni[1,2], Franziska Kundel[1], Tuomas P.J. Knowles[1,2,4], Christopher M. Dobson [1,2], Clare Bryant[3], Michele Vendruscolo [1,2] & David Klenerman[1,5]

Protein aggregation is a complex process resulting in the formation of heterogeneous mixtures of aggregate populations that are closely linked to neurodegenerative conditions, such as Alzheimer's disease. Here, we find that soluble aggregates formed at different stages of the aggregation process of amyloid beta (Aβ42) induce the disruption of lipid bilayers and an inflammatory response to different extents. Further, by using gradient ultracentrifugation assay, we show that the smaller aggregates are those most potent at inducing membrane permeability and most effectively inhibited by antibodies binding to the C-terminal region of Aβ42. By contrast, we find that the larger soluble aggregates are those most effective at causing an inflammatory response in microglia cells and more effectively inhibited by antibodies targeting the N-terminal region of Aβ42. These findings suggest that different toxic mechanisms driven by different soluble aggregated species of Aβ42 may contribute to the onset and progression of Alzheimer's disease.

[1] Department of Chemistry, University of Cambridge, Lensfield Road, Cambridge CB2 1EW, UK. [2] Centre for Misfolding Diseases, Department of Chemistry, University of Cambridge, Lensfield Road, Cambridge CB2 1EW, UK. [3] Department of Veterinary Medicine, University of Cambridge, Madingley Road, Cambridge CB3 0ES, UK. [4] Cavendish Laboratory, University of Cambridge, J J Thomson Avenue, Cambridge CB3 1HE, UK. [5] UK Dementia Research Institute, University of Cambridge, Cambridge CB2 0XY, UK. These authors contributed equally: Suman De, David C. Wirthensohn, Patrick Flagmeier, Craig Hughes, Francesco A. Aprile, Francesco S. Ruggeri. Correspondence and requests for materials should be addressed to S.D. (email: sd730@cam.ac.uk) or to M.V. (email: mv245@cam.ac.uk) or to D.K. (email: dk10012@cam.ac.uk)

M any human pathologies, including neurodegenerative disorders such as Alzheimer's disease (AD), are associated with the aggregation of amyloidogenic proteins[1,2]. In AD, the normally soluble amyloid beta peptide (Aβ) forms insoluble deposits in the brains of affected individuals[3,4]. The molecular mechanism that governs the aggregation of Aβ involves the formation of a variety of soluble oligomeric intermediates of different sizes and structures[5,6]. It has been shown that these soluble oligomeric species play a critical role in the cascade of events that ultimately leads to AD via a wide range of neurotoxic events, including the permeabilization of cellular membranes, calcium dysregulation, mitochondrial damage, induction of inflammation, and oxidative stress[7–11].

The soluble oligomeric aggregates formed during the process of amyloid formation are typically highly heterogeneous, and exist in a variety of sizes, shapes and structures, with different degrees of β-sheet content and surface hydrophobicity[11,12]. These differences could give rise to variations in both the mechanism and degree of cytotoxicity. It is therefore important to characterize the structural differences between different soluble aggregates formed over the course of the aggregation reaction and to determine how specific structural features are related to toxicity and could contribute to disease pathogenesis. These soluble aggregates populated during the aggregation of Aβ in particular have been shown to be low in abundance and heterogeneous in both size and conformation, making it challenging to characterize their structure and mechanism of toxicity[13]. In this context, a recent advance has been the design and generation of rationally designed antibodies that recognize specific epitopes of Aβ and inhibit its aggregation[14,15]. The binding of such antibodies to Aβ aggregates provides information about the accessibility of specific epitopes, and have been found to inhibit aggregate-induced toxicity[11]. Thus, the systematic targeting of soluble aggregates of Aβ with designed antibodies should allow us to determine the accessibility and significance of individual regions of the Aβ sequence in generating aggregate-induced toxicity.

Given the heterogeneous nature of these soluble aggregates, we use a previously described gradient ultracentrifugation method that allows oligomeric species formed in an aggregation reaction to be separated on the basis of their size and density[16–18]. We study the aggregation of Aβ42, the 42-residue isoform of Aβ, and monitor the toxicity for aggregates of different sizes in the presence of the designed antibodies that target distinct epitopes. Two ultrasensitive methods able to measure well-defined effects of the aggregates are used. The first is a quantitative assay that determines the ability of the protein aggregates to induce the permeabilization of lipid bilayers and the second is an assay that measures the ability of aggregates to induce inflammation through the measurement of cytokine production by microglial cells. We observe a relatively decreased ability of the oligomers to disrupt lipid bilayers and a relatively increased capability to cause inflammation as the size of the Aβ42 aggregates increased. We then find that these effects correlate with changes in the surface structure of the aggregates as indicated by differences in the effects of the designed antibodies.

## Results
**Different Aβ42 aggregates have distinct toxicity mechanisms.** To test quantitatively the level of lipid membrane permeability induced by soluble Aβ42 aggregates formed at different stages of the aggregation reaction (Fig. 1a), we used a recently developed high-throughput assay based on the measurement of changes in the localized fluorescence intensity of individual nano-sized lipid vesicles[19]. We immobilized thousands of individual vesicles onto glass cover slides, which act as individual nano-reactors,

each containing a reporter of the presence of calcium ions ($Ca^{2+}$) that enter into the vesicle, the quantity of which can be measured using total internal reflection fluorescence (TIRF) microscopy[19–22] (Fig. 1b). We have previously shown that monomeric Aβ42 does not induce membrane permeability and that soluble aggregates rather than mature fibrils are the species responsible for inducing this membrane disruption and ion influx[19]. For this study, we aggregated monomeric Aβ42 at a concentration of 2 μM under quiescent conditions and removed aliquots from the solution at different time points. Using this assay, we found that aliquots taken at an early time point (15 min) in the aggregation reaction, when diluted to a concentration of 50 nM, induced twice as much $Ca^{2+}$ influx compared to aggregates present in the aliquots taken at later stages (30 min) of the aggregation reaction (Fig. 1b, c).

By contrast, we observed an opposite trend in the effects of soluble Aβ42 aggregates when monitored by an inflammation assay. Inflammation is driven by pattern recognition receptors, such as toll-like receptors (TLRs)[23,24]. Multiple studies have demonstrated that soluble forms of Aβ interact with the toll-like receptor 4 (TLR4) on microglia cells, resulting in the production of inflammatory cytokines that are thought to contribute to the neuronal damage associated with the progression of AD[24–27]. To study and quantify the inflammation induced in cells by the Aβ42 protein aggregates, we monitored the levels of secreted tumour necrosis factor alpha (TNF-α), a pro-inflammatory cytokine, using an enzyme-linked immunosorbent assay (ELISA) assay[28] (Fig. 1d). We took aliquots from the same aggregation reaction of Aβ42 as described above, diluted them to monomer-equivalent concentrations of 1 μM, added the samples to BV2 microglia cells, and incubated for 24 h before measuring the cumulative TNF-α production from the supernatant, using the ELISA assay. We found that addition of the aliquots of the Aβ42 aggregation reaction mixture did indeed activate an innate immune response through the production of significant amounts of TNF-α; by contrast, addition of samples containing monomeric or fibrillar forms of Aβ42 did not cause any significant inflammatory response (Supplementary Fig. 1a). We also observed that the inflammatory response for the oligomeric aggregates could be effectively blocked with 100 ng/mL of the TLR4 antagonist *Rhodobacter sphaeroides* lipid A (RSLA) (Supplementary Fig. 1b), indicating that TLR4 signalling plays a dominant role in the observed response. We observed that aggregates present in aliquots removed at the later stage (30 min) in the Aβ42 aggregation process induce a stronger inflammatory response than the aggregates present in the aliquots taken at the early stage (15 min) of the aggregation reaction (Fig. 1e).

**Size and toxicity mechanism of Aβ42 aggregates are linked.** The differences between the species of Aβ42 in the aliquots taken from the aggregation reaction at different times in the two toxicity assays used in this work could be due to differences in the size and structure of aggregates since both of these properties change over the course of the aggregation reaction. To test this hypothesis, we employed a sucrose gradient ultracentrifugation technique by means of which protein aggregates of different sizes can be separated into distinct fractions[13,16,18]. Therefore, we combined in equal proportions aggregates taken from three different time points during the aggregation reaction; (i) the end of the lag phase, (ii) the middle of the growth phase and (iii) in the plateau phase (Fig. 2a). This mixture was injected over a sucrose gradient prepared by adding 600 μL of 10%, 20%, 30%, 40% and 50% (w/v) sucrose solutions from top to bottom in a rotor tube and ultracentrifugation was performed (Fig. 2b). This technique allowed us to separate the soluble aggregates of Aβ42 into different fractions,

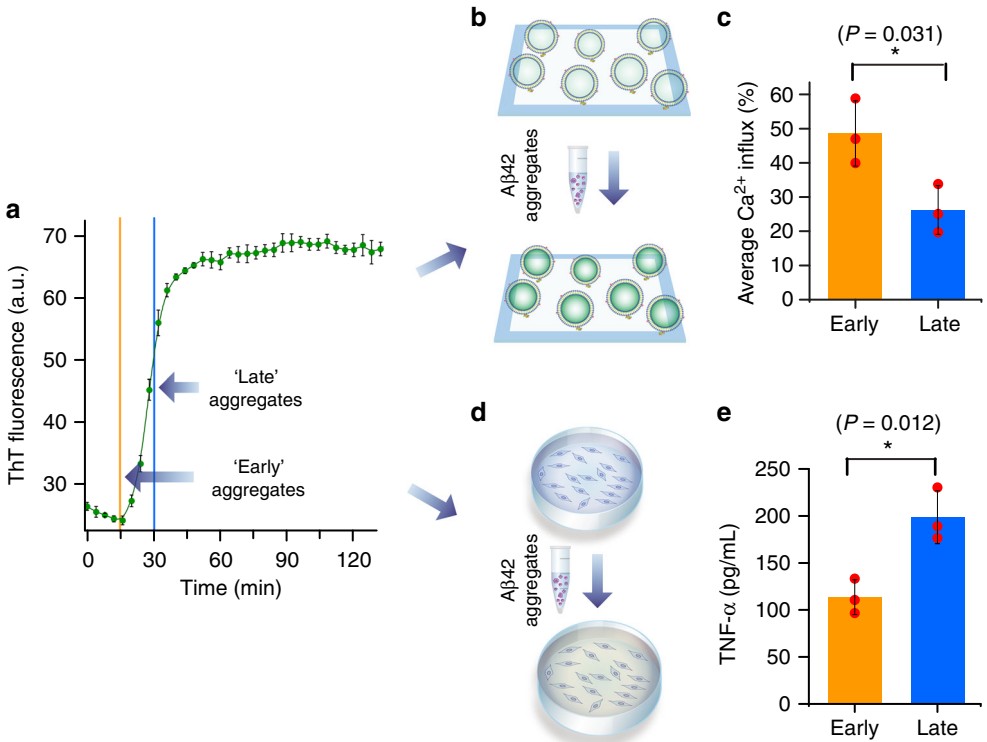

**Fig. 1** Soluble Aβ42 aggregates at different aggregation stages show different toxicity mechanisms. **a** The aggregation Aβ42 was monitored by ThT fluorescence. Each data point represents the mean of three independent biological repeats and its error bar the corresponding standard deviation. Aliquots containing Aβ42 aggregates were taken from the aggregation reaction at two different time points—at the end of the lag phase (15 min, for 'early' soluble aggregates) and at the midpoint of the growth phase (30 min, for 'late' soluble aggregates). **b** The lipid bilayer permeability induced by Aβ42 aggregates was measured by immobilizing on glass cover slides hundreds of nano-sized individual lipid vesicles in the presence of a $Ca^{2+}$-sensitive dye. If the aggregates disrupt the integrity of the lipid bilayer, $Ca^{2+}$ from the external medium enters into individual vesicles to a level that can be quantified using highly sensitive total internal fluorescence microscopy (TIRF) measurements. **c** Early Aβ42 aggregates cause a higher level of bilayer permeability than later Aβ42 aggregates. The data were averaged over three biological repeats, with each repeat involving the analysis of 6000 individual lipid vesicles, the error bars represent standard deviation, and the statistical significance was calculated using a two-sample $t$-test. **d** The inflammatory response in microglia was quantified using an ELISA assay to measure the levels of secreted tumour necrosis factor alpha (TNF-α). **e** In contrast to the bilayer permeability assay, the late aggregates were more effective than the early aggregates. The data are averaged over three biological repeats, each repeat was averaged over three wells, each of which contained 200,000 to 300,000 BV2 cells, the error bars represent the standard deviation among the repeats, and the statistical significance was calculated using a two-sample $t$-test (unpaired). Data points for each biological replicate shown with red circles. Source data of **a**, **c** and **e** are provided as Source Data files. Aggregates used for the membrane permeability and the inflammation assays were collected from the same aggregation reaction

based on their mass, for testing in two toxicity assays. We found that the aggregates present in the 20% sucrose solution induced $Ca^{2+}$ influx most effectively (Fig. 2c), and that those present in the 30% fraction were most effective at inducing an inflammatory response (Fig. 2d). The supernatant, pallets and sucrose do not induce any inflammatory response or membrane permeation (Supplementary Fig. 2).

To characterize the aggregates present in the 20% and 30% sucrose solutions, we used high-resolution and phase-controlled atomic force microscopy (AFM) to acquire 3D images of the aggregates within each fraction with sub-nanometre resolution[29]. Remarkably, this approach is capable to acquire the morphology of biomolecules as small as low molecular weight monomers (Aβ42, ~4 kDa)[30]. Then, by means of statistical analysis of the cross-sectional dimensions of the species in the individual images[31], is capable to distinguish monomers from the early oligomeric aggregated species, a capability that is at the limit of any existing biophysical single-molecule method. In Fig. 3a–e, we show typical 3d morphology maps of the 10%, 20%, 30%, 40% and 50% fractions (left panels) along with high-resolution images of typical species present in each fraction (central panel) with the relative cross-sectional height (right panel). In the right

column of Fig. 3 we show that the results of the statistical analysis of the cross-sectional dimensions of the aggregates within each fraction. While the measurement of the cross-sectional height of the aggregates can be measured with angstrom resolution, the measurement of cross-sectional diameters is limited by the AFM tip size and the consequent convolution effects[32]; we therefore used an established approach to obtain de-convoluted cross-sectional diameters of the aggregates[33,34] (see Methods for details). The results showed a significant increase in the average height of the aggregates as a function of the sucrose fraction density (Fig. 4a); an increase in the cross-sectional diameter was also visible for the 40% and 50% fractions (Fig. 4b). The smallest observed cross-sectional height (0.3–0.5 nm) in all the fractions corresponded to the size of the monomeric protein. The average level of noise is well below 0.1 nm and the signal to noise ratio for measuring a single monomer is approximately 10, which allows the accurate determination of cross-sectional height of individual aggregate (Supplementary Fig. 3). The 10% and 20% fractions showed a high abundance of both monomeric and approximately spherical oligomeric species (the latter with heights of 0.5–2 nm)[31,34]. The 30% sucrose fraction showed a smaller number of monomeric species and the presence of both

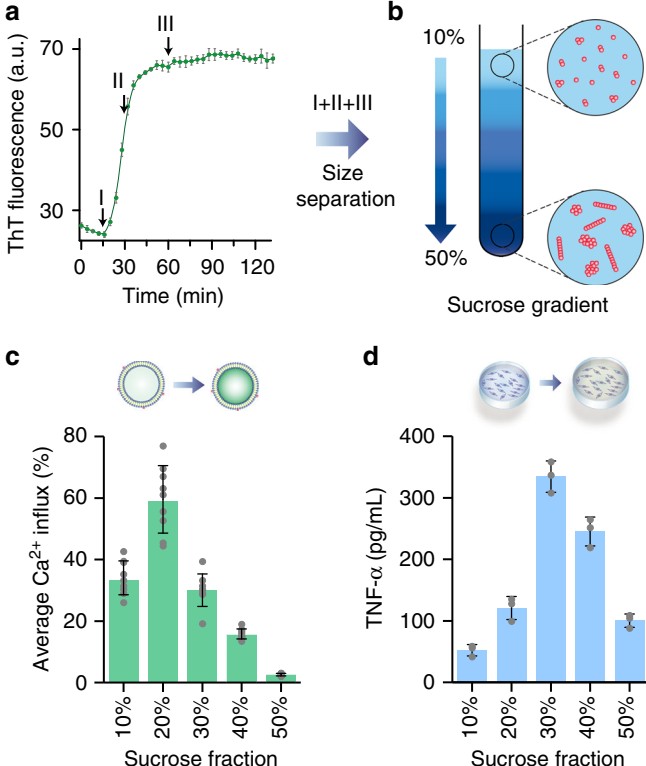

**Fig. 2** Aβ42 aggregates of diverse sizes exhibit different relative toxicity by distinct mechanisms. **a** Aβ42 aggregation monitored by ThT fluorescence. Soluble aggregates formed at different time points were mixed together and separated via a discontinuous sucrose gradient. Three different time points during the aggregation of 2 μM Aβ42 in SSPE buffer at 37 °C were chosen: (i) At the end of the lag phase (15 min), (ii) at the middle of growth phase (30 min) and (iii) during the plateau phase (60 min). **b** Aliquots collected at these time points were loaded in a step gradient of 10% to 50% sucrose and ultracentrifuged and the fractions were collected immediately and stored. The sizes of the aggregates present within different densities of the sucrose solution increase with the sucrose density. **c** The lipid bilayer permeability assay shows that aggregates present at 20% sucrose are the most potent at membrane permeation. **d** Aggregates presents at 30% sucrose are the most effective at inducing inflammation. These experiments were carried out for two independent aggregation reactions of Aβ42 ($n = 2$) and the error bars represent the standard deviation of the mean for each field of view (for 2C) and for each well (2D). Source data of **a**, **c** and **d** are provided as a Source Data file

spherical oligomers and more elongated filamentous species, the latter being identified as protofilaments with a mean height of 0.4–1 nm[31] and a length of several hundred nm. The 30% sucrose fraction, showing the highest increase in TNFα levels, was the only one in which such protofilaments were visible. The fractions at higher percentages of sucrose, 40% and 50%, showed only higher order aggregates.

We then characterized the structures of the individual aggregates present in different sucrose fractions using a single-particle Förster resonance energy transfer (FRET) approach[35,36]. First, we generated co-aggregates from the 1:1 molar ratio of Aβ42 by aggregating monomers labelled with FRET donors and FRET acceptors (Fig. 5a, b). This approach allowed us to define the relative aggregate sizes and their FRET efficiencies as described previously[37–39]. We observed a gradual increase in the FRET efficiency with increasing aggregate size, which is indicative that the donors and acceptors are increasingly close together in the larger oligomers (Fig. 5c and Supplementary

Fig. 4). Second, we used total internal reflection (TIRF) microscopy to observe the aggregates both with the FRET samples and using extrinsically fluorescent dyes such as thioflavin-T (ThT) and pFTAA (Fig. 5d and Supplementary Fig. 5)[40,41]. This strategy allowed us to determine the fluorescence intensities of individual aggregates, and the data show a gradual increase in mean intensity from low to high sucrose fractions (Fig. 5e, f), i.e. with an increase in aggregate size. This approach was particularly useful for sub-diffraction sized aggregates that appear to have similar diameters but to differ in fluorescence intensity, while in the higher density sucrose fractions the morphology of the fluorescence spots was observed to change from diffraction limited points to elongated fibrils.

In combination these data show that the smaller aggregates present in low sucrose density solutions are on average more potent at membrane permeation than the relatively larger aggregates present in high sucrose density solutions. Since the ability of protein aggregates to permeabilise lipid membrane can be correlated with their surface hydrophobicity, we have performed an advanced spectrally resolved super-resolution imaging to map the surface hydrophobicity of soluble Aβ42 aggregates at the single-aggregate level[42,43]. We used a solvato-chromic dye, Nile Red, which transiently binds to protein aggregates and whose emission spectra is sensitive to the local hydrophobicity[42,43]. We compared the average of the mean emission maxima from individual aggregates and found that the 20% faction was more hydrophobic than the aggregates present at higher sucrose fractions (Supplementary Fig. 6) with a blueshift of approximately 10 nm compared to the aggregate present at 30% and 40% sucrose fraction. This observation is consistent with previous observations that the ability to permeabilize membranes is higher for small aggregates than for larger ones, and is enhanced by the increased exposure of hydrophobic patches[44,45]. In addition, the small soluble species present at early stages of the aggregation process have been reported to have higher tendencies to interact, permeabilize and cross the plasma membrane than larger aggregates[44]. Interestingly, the larger aggregates, which appear elongated and similar to descriptions of protofilaments[31], that are present in 30% sucrose solution were found to be more effective at inducing inflammation.

**Roles of Aβ42 sequence regions in the toxicity mechanisms.** We next used a panel of single-domain antibodies to probe the regions of the Aβ42 peptide that are important for generating the membrane permeability and inflammatory response induced by the different oligomeric aggregates. We employed five rationally designed antibodies (DesAbs) that target five different linear epitopes on Aβ42 that effectively scan its whole sequence[15]. In particular, the third complementary-determining region (CDR3) of antibody DesAb3–9 was grafted with a sequence (HETLTLR), which is designed to bind to the N-terminal region of Aβ42. The antibodies DesAb13–19 (CDR3: LSVIKEI) and DesAb18–24 (CDR3: VFVGTEA) were designed to bind to the central region of Aβ42, and the antibodies DesAb29–36 (CDR3: GSMYKATV) and DesAb36–42 (CDR3: LGIKAEL) were designed to target its C-terminal region of (Fig. 6a). These DesAbs have been found to bind with much higher affinity to aggregates of Aβ42 than to the corresponding monomeric species[13].

We first assessed the ability of the designed antibodies to inhibit the Aβ42 aggregate-induced membrane permeability. Different concentrations of the DesAbs were added to aliquots of the solutions from the aggregation reaction described above containing the differently sized species present in the 20% and 30% sucrose fractions. We first measured the ability of these species to disrupt lipid membranes by measuring the extent of

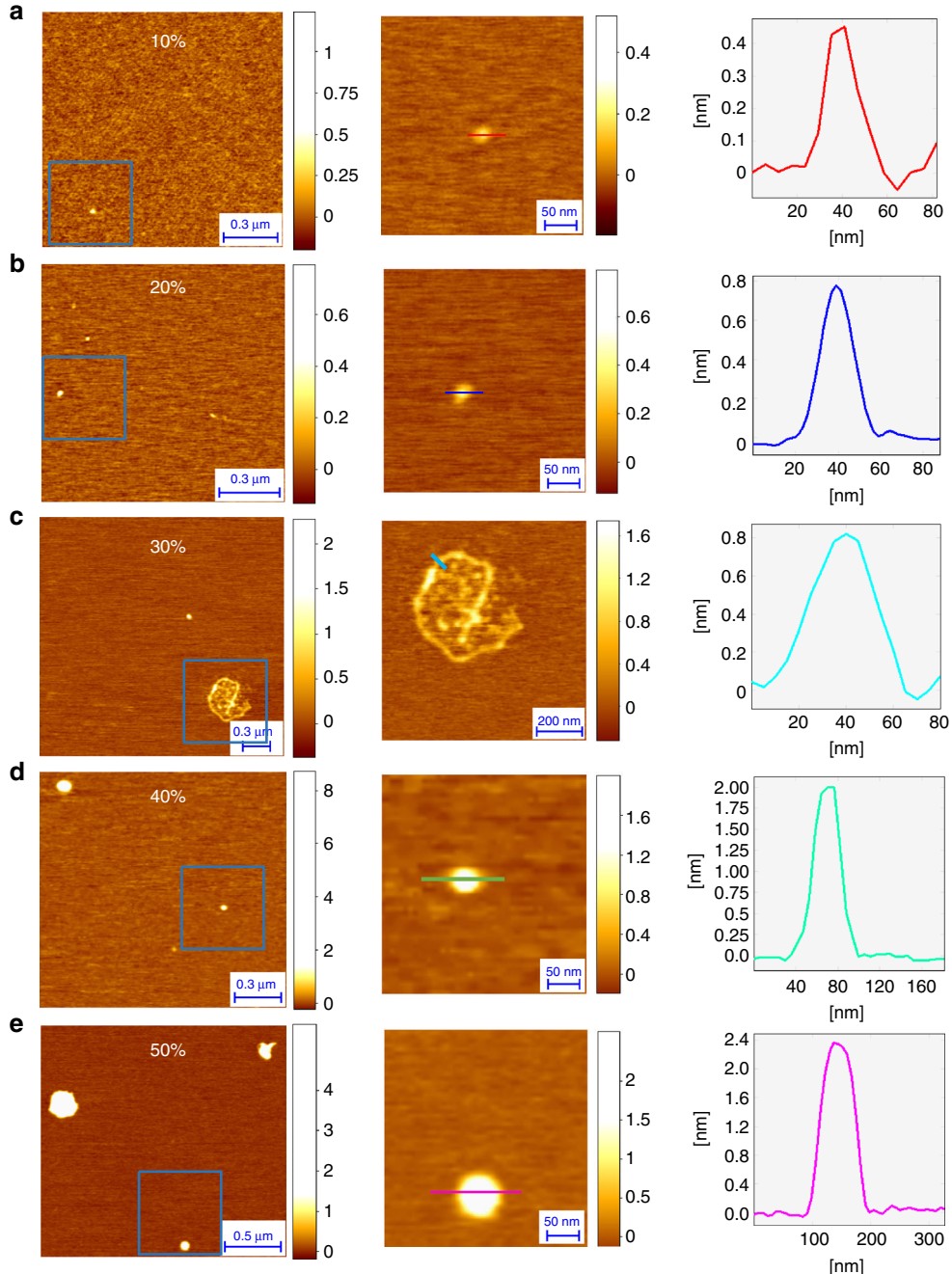

**Fig. 3** AFM characterization of aggregates present in sucrose fractions. **a–e** Representative AFM images of the Aβ42 aggregates present at different fractions of sucrose gradient (left panels), with a magnified example of the species present in each fraction (central panel) and the corresponding cross-sectional dimensions (right panel). The 30% fraction (**c**) is the only one that shows protofilaments aggregates

$Ca^{2+}$ influx into the tethered vesicles (Fig. 6b, c). We observed that all the DesAbs reduced the aggregate-induced membrane permeability, and that the ability to do so increased progressively as the epitope on Aβ42 varied from the N-terminal to the C-terminal region. Indeed, DesAb36–42, which binds to the most C-terminal region of Aβ42, was found to be the most effective antibody in suppressing the induction of bilayer permeability. This result suggests that the hydrophobic C-terminal region of Aβ42 is directly involved in the mechanism by which perme-ability occurs. The concentration required for all the antibodies, including DesAb36–42, to reduce the $Ca^{2+}$ influx by 50% was similar for the aliquots taken from both the 20% and 30% sucrose fractions, suggesting that the structural characteristics enabling

the aggregates to induce bilayer permeability are not significantly affected by their overall size. In addition single-particle imaging measurements of the populations of Aβ42 aggregates able to bind to ThT-active species[41] showed no significant difference in the number of such species detected before and after incubation with the antibodies, indicating that none of the panel of antibodies disrupted the aggregates detectably within the time frame of our experiments (Supplementary Fig. 7).

We next examined the ability of the same panel of antibodies on the inflammation induced by Aβ42 aggregates. The aggregates contained in the 20% and 30% sucrose fractions were incubated with varying concentrations of each antibody, and then added to BV2 microglial cells. Although addition of antibodies resulted

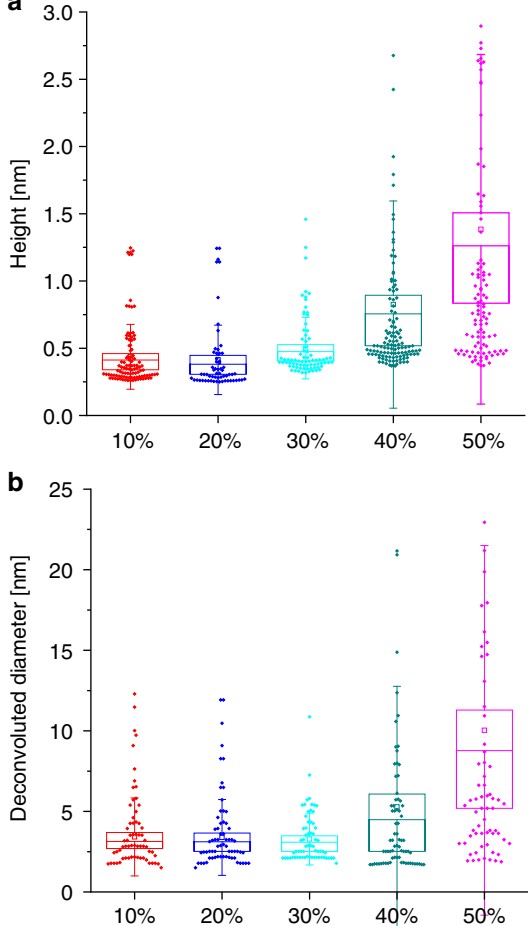

**Fig. 4** Statistical analysis of aggregates present in different sucrose fractions. As represented by box charts of the **a** height and **b** the deconvoluted diameter of the Aβ42 aggregates are shown. As the concentration of sucrose increase, the height as well as the diameter of individual aggregate increases. Source data of **a** and **b** are provided as a Source Data file

in a decrease in TNF-α expression, we observed that much higher concentrations of antibodies were needed to reduce inflammation than to decrease lipid permeability. Moreover, in contrast to the observation that the inhibition of bilayer permeability was greatest with the antibody DesAb36–42, the most significant reduction in the levels of TNF-α expression was found with the antibody DesAb3–9, which binds close to the N-terminus of the peptide. We also observed a general decrease in the efficacy of the antibodies as the binding epitope is moved closer to the C-terminal region of Aβ42 (Fig. 6d, e). As with the permeability assay, we observed the same fractional reduction in TNF-α production with each antibody for aggregates of different sizes, showing that the regions of the aggregates responsible for inducing inflammation are structurally accessible in the same way in the larger and smaller species. It has to be noted that these antibodies do not induce any significant membrane permeation or inflammation (Supplementary Fig. 2).

## Discussion
We have shown that soluble Aβ42 aggregates of different sizes vary in their relative contributions to different mechanisms of toxicity, namely the induction of membrane permeability and of inflammation. The smaller aggregates formed at an early stage of the aggregation process of Aβ42 were observed to be more

effective at inducing permeability of lipid bilayers, whereas larger aggregates formed at a later stage of the aggregation reaction were more potent at inducing an inflammatory response. Using rationally designed single-domain antibodies, we employed an epitope scanning technique to determine which regions of the sequence of Aβ42 are most solvent accessible in the two types of aggregates. We observed that all the antibodies inhibited both types of toxicity but that those that bind to the C-terminal region of Aβ42 were more effective in counteracting the induction of bilayer permeability than the antibodies that bind to the N-terminal region. We observed an opposite trend, however, for the inhibition of aggregate-induced inflammation in microglia cells, where antibodies designed to bind to the N-terminal region of the sequence were more effective. These differences are likely to be due to the favourable accessibility of solvent exposed N-terminal segments in the larger aggregates of Aβ42 that are formed at later stages of the aggregation process.

The structures of Aβ42 fibrils determined using cryo-electron microscopy and nuclear magnetic resonance spectroscopy show that the N-terminal residues are substantially exposed within the assemblies, whereas the C-terminal residues are largely located within the hydrophobic core of the cross-β structure[46,47]. Our observation that DesAb3–9, which binds to the N-terminal region, is the most effective in counteracting the toxicity of the various Aβ42 aggregates indicates that this region is similarly exposed in the soluble oligomeric aggregate structures[47], and indeed that the N-terminal residues are important for TLR signalling and hence for inflammation[23]. Moreover, larger protofibrils or protofilaments are likely to present an increased number of TLR4 binding sites on their surfaces, the majority of which are likely to need to interact with the antibodies to block signalling. This scenario would provide an explanation of why much higher concentrations of designed antibodies were found to be needed to reduce the inflammatory response than for suppressing the degree of bilayer permeability experiments. Moreover, the ability of the antibodies that bind to the C-terminal region to counter the bilayer permeability, which is induced most effectively by the smaller Aβ42 aggregates, suggests that the hydrophobic C-terminal region of these oligomeric species is exposed in this species, and responsible for the observed disruption of the lipid bilayer.

Our results show that structurally different aggregated species exist within the heterogeneous ensemble of soluble Aβ42 aggregates and that these can exert toxicity by distinct mechanisms. As the aggregation process proceeds, and the Aβ42 aggregates on average increase in size and undergo structural changes, as we have previously observed with α-synuclein[39,48], their ability to disrupt the integrity of lipid bilayers of the type involved in cellular membranes appears to be reduced relative to triggering an inflammatory response. These results suggest that different aggregated forms of Aβ42 may differ in their contributions to different mechanisms of toxicity that develop during the progression of AD and other misfolding disorders associate with Aβ self-assembly. We therefore anticipate that effective strategies aimed at targeting pathogenic Aβ42 species formed during the progression of disease may involve the use of cocktails of therapeutic agents targeting the diversity of species that are populated during the aggregation process.

## Methods
**Purification and aggregation of synthetic Aβ42.** The Aβ42 peptide (DAEFRHDSGYEVHHQKLVFFAEDVGSNKGAIIGLMVGGVVIA) (Abcam) was purified with slight modifications[18]. Solutions of monomeric Aβ42 were prepared by dissolving the lyophilized Aβ42 peptide in 6 M GuHCl and then purifying the peptide using a Superdex 75 10/300 GL column (GE Healthcare Bio-Sciences AB SE-751 84). The centre of the elution peak was collected, and the peptide concentration was determined from the absorbance of the integrated peak area

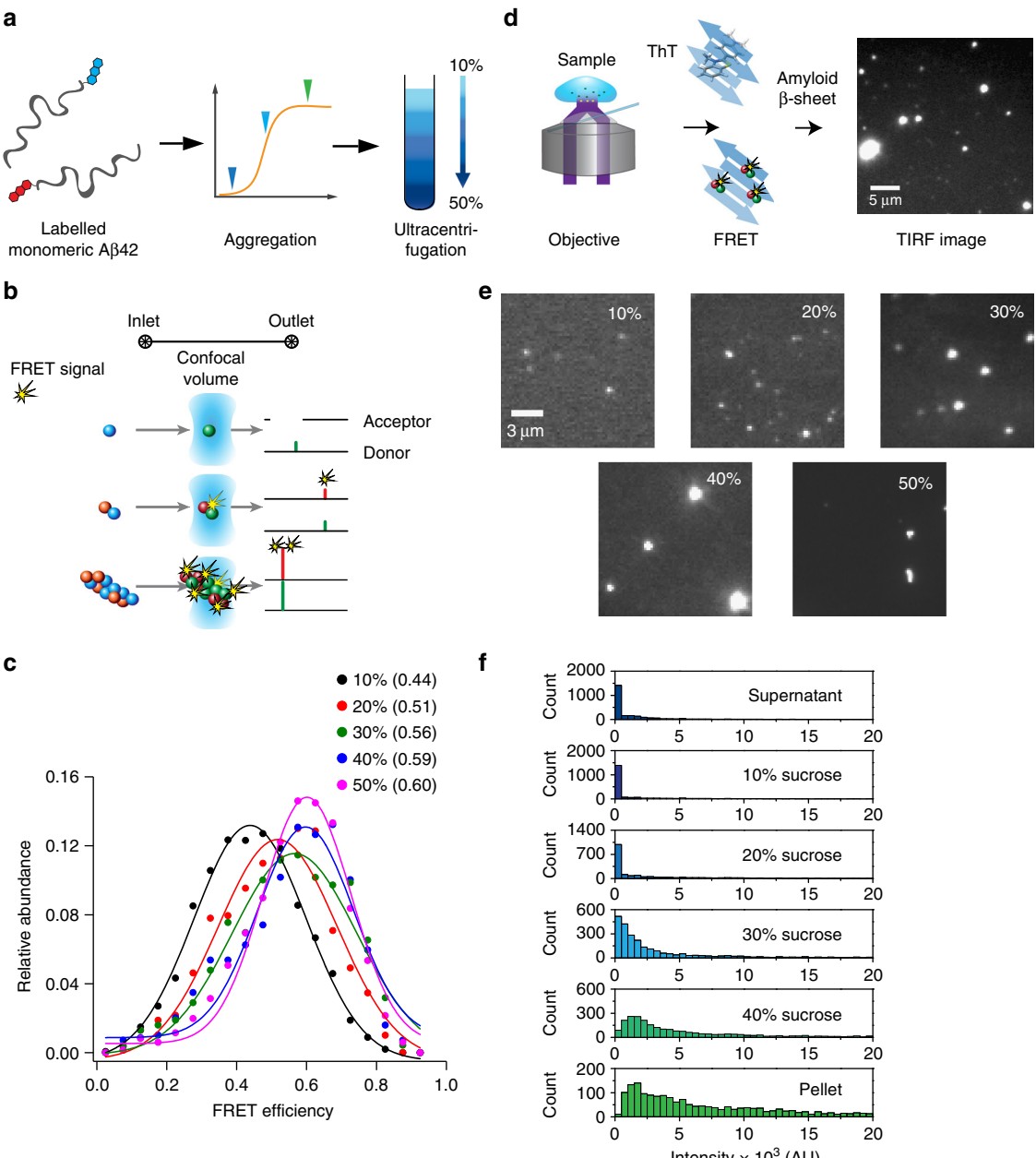

**Fig. 5** Single-particle characterization of the dye-labelled Aß42 aggregates. **a** Schematic representation of the separation workflow of the Aβ42 aggregates: solutions of dye-labelled monomeric Aβ42 were allowed to aggregate and aliquots removed at the different times described in Fig. 2 (shown as coloured triangles in the graph). The aggregated mixture collected at different time points was then loaded onto a sucrose step gradient that is divided into five fractions, and was subsequently used for experimental measurements. **b** Illustration of the confocal FRET experiment, showing a schematic of the microfluidic channel used to deliver the samples for measurement. The confocal volume of the excitation beam was set to the centre of the channel. Both the donor and acceptor channel intensities were recorded for every aggregate flowing through the excitation volume. **c** Maxima of FRET efficiency for increasing sucrose concentrations shifts towards higher FRET efficiencies indicates that size of the aggregates present in sucrose solution increases. Relative FRET efficiency for each sucrose concentration are fitted with Gaussian function (solid lines) to get the maxima (shown in parenthesis). **d** Schematic of the TIRF imaging for single-aggregate imaging. **e** Images of aggregates present in all five fractions are shown to represent the sizes detected through TIRF microscopy and are representative of the indicated fractions. The scale bar is 3 μm. The FRET histograms of the fractions show a clear shift from a very narrow, low-intensity distribution in the smaller fractions to a broader higher intensity distribution in the higher fractions and in the pellet. Each histogram is the sum of aggregates of 27 field of views measured in three repeats of a single gradient ultracentrifugation preparation. Source data of **c** and **e** are provided as a Source Data file

using $\varepsilon_{280} = 1490$ L/molcm. Aliquots of monomeric Aβ42 were diluted with SSPE buffer to a concentration of 2 μM in low-binding Eppendorf tubes on ice. Individual samples were then pipetted into multiple wells of a 96-well half-area plate (Corning 3881) and the plate was placed into an incubator at 37 °C, under quiescent conditions. Aliquots for measurements were then taken at ~15 min (early aggregates) and ~30 min (late aggregates) after the plate was placed in the incubator.

**Single vesicle based membrane permeabilization assay.** Vesicles with average diameter of 200 nm were prepared by five freeze and thaw cycles combined with extrusion as previously described[19]. Briefly, 16:0–18:1 PC (10 mg/mL) and 18:1–12:0 Biotin PC (1 mg/mL) (Avanti Lipids) were mixed with 100:1 ratio. The lipid mixture was then hydrated in HEPES buffer (50 mM, pH 6.5) with 100 μM Cal-520 (Stretch). Freeze and thaw cycles were performed using a water bath and dry ice to get the unilamellarity. The lipid solution was extruded at least for 10

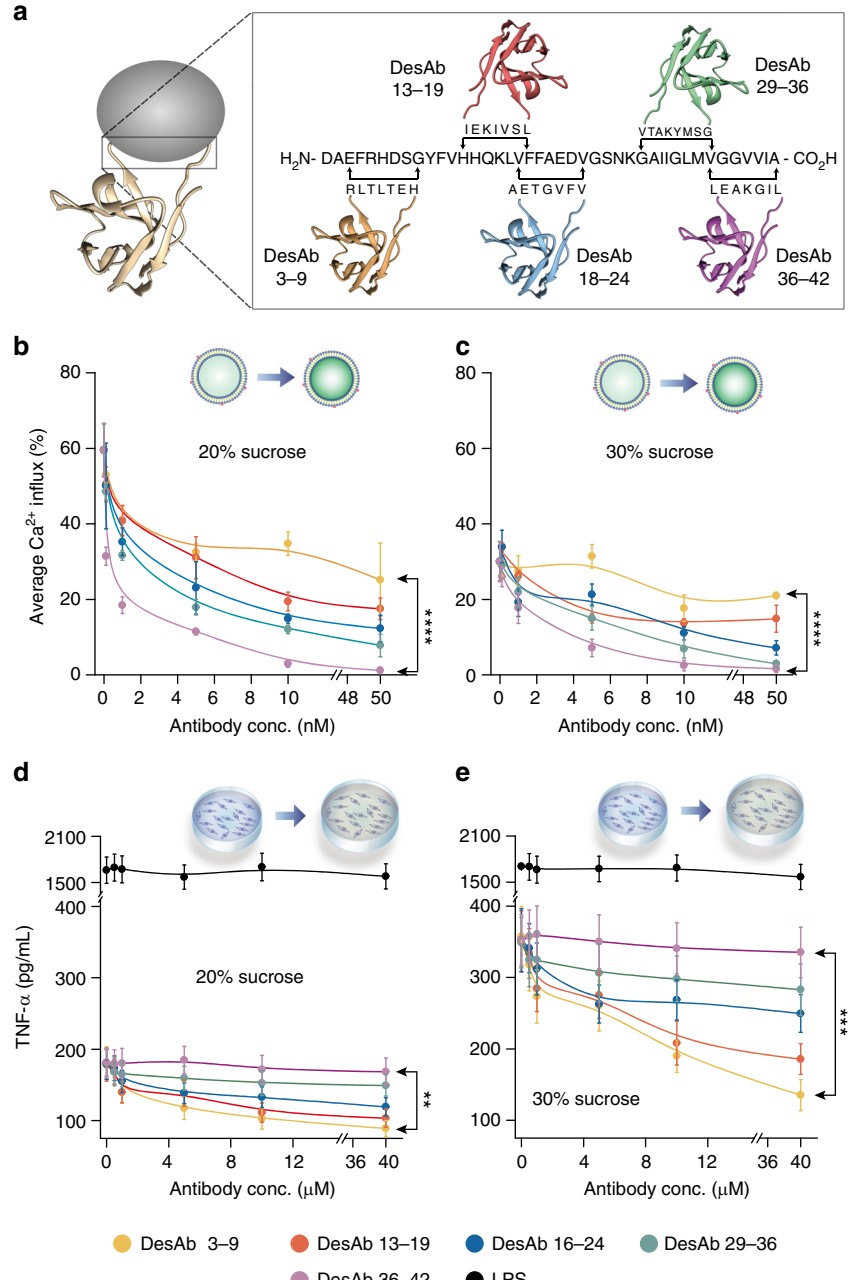

**Fig. 6** Influence of the different sequence regions of Aβ42 on its toxicity mechanisms. **a** We used five rationally designed antibodies that target different epitopes of the Aβ42 sequence. The antibodies DesAb$_{3-9}$ and DesAb$_{36-42}$ bind to the N-terminal and C-terminal regions of Aβ42, respectively. Representative experiments showing the concentration of each antibody (x-axis) added to aggregates present in solutions at **b** and **d** 20% and **c** and **e** 30% sucrose and the reduction of aggregate-induced toxicity. The error bars represent the standard deviation among the field of views (**b**, **c**) and among the well (**d**, **e**). Antibodies that target C-terminal regions of Aβ42 are more effective at reducing the membrane permeability induced by Aβ42 aggregates. Antibodies that target N-terminal regions of Aβ42 are more potent in reducing the inflammatory response induced by Aβ42 aggregates (**d**, **e**). As a positive control, we used lipopolysaccharides (LPS), which is known to induce TNF-α production in microglial cells. For each case, P values are calculated using two-sample t-test to compare the inhibition by most N-terminally binding antibody (DesAb3–9) and C-terminally antibody (DesAb36–42) at their highest concentration (two biological repeats n = 2, the lines are simply guides to the eye). Source data of **b**–**e** are provided as a Source Data file

times through a membrane with a size cut off of 200 nm and size of the vesicles were measure using a Zetasizer (Zetasizer Nano ZSP). Free dyes were removed using size exclusion chromatography.

The single vesicle based assay was used to measure the permeability of lipid bilayers in presence of toxic protein aggregates[19,49]. Glass cover slides (VWR International, product number 63 1-0122) were cleaned by sonicating in 2% (v/v) Hellmanex III (Hellma GmbH & Co. KG) in milliQ water for 10 min followed by sonicating twice in milliQ water and in methanol for 10 min each. Then the cover slides were dried under a stream of nitrogen gas, and plasma-etched using an argon plasma cleaner (PDC-002, Harrick Plasma) for at least 30 min to remove any

fluorescent impurities. Each cover slide was fixed by frame-seal incubation chambers (Biorad, Hercules) and the surface was coated with 100:1 PLL-g-PEG and PLL-g-PEG biotin (SuSoS AG) (1 g/L) in 50 mM HEPES buffer. Then the cover slides were washed three times and 0.1 mg/mL neutravidin (Thermo Scientific) solution was added to the cover slide and incubated for 15 min and washed three times with reaction buffer. Then, a 50 μL aliquot of the solution of purified biotinylated vesicles was added to the cover slide and incubated for 30 min before washing carefully at least five times with reaction buffer. Single vesicles tethered to borosilicate glass cover slides via biotin neutravidin linkage were incubated with 50 μL Ca²⁺ containing buffer solution Leibovitz's L-15 (phenol red free) and

background image was recorded ($F_{background}$). Thereafter, 50 μL of sample was added and images were acquired ($F_{sample}$) and care was taken to avoid moving the glass cover slides during the addition of each sample. Next, 10 μL of a solution containing 1 mg/mL of ionomycin, an ionophore for $Ca^{2+}$ ion, and subsequently images of $Ca^{2+}$ saturated single vesicles in the same fields of view were acquired ($F_{Ionomycin}$). For each field of view 50 images were taken with an exposure time of 50 ms. The fields of view were chosen using an automated programme (homemade bean-shell script, Micromanager) to avoid any user bias, which can be available upon reasonable request. The relative influx of $Ca^{2+}$ into an individual vesicle due to aggregates of Aβ42 was then determined as

$$Ca^{2+} influx = \frac{F_{sample} - F_{background}}{F_{Ionomycin} - F_{background}}. \quad (1)$$

The average degree of was calculated by averaging the $Ca^{2+}$ influx into individual vesicles. Where representative data set is shown, all data sets show similar qualitative trends. The sucrose fractions were diluted 1:10 for all experiments in order to be in the linear regime of the measurement.

**TIRF microscopy**. Imaging was performed using a homebuilt TIRFM microscope based on an inverted Nikon Ti microscope using a 488 nm laser for membrane permeabilization assay[35]. The expanded and collimated laser beams were focused onto the back-focal plane of the ×60, 1.49 NA oil immersion objective lens (APON60XO TIRF, Olympus, product number N2709400). The fluorescence signal emerged from the sample was collected by the same objective and was separated from the excitation beam by a dichroic (Di01-R405/488/561/635). The emitted light was passed through an appropriate set of filters (for single vesicle assay BLP01-488R, Semrock and FF01-520/44-25) to remove any unwanted light. The fluorescence signal was then expanded 2.5 times and imaged onto a 512 × 512 pixel EMCCD camera (Photometrics Evolve, E VO-512-M-FW- 16-AC-110). The open source software Micro Manager 1.4 was used to control the microscope hardware and image acquisition[50]. Images were acquired in an unbiased way using bean-shell based script.

**Hydrophobicity mapping of individual Aβ42 aggregates**. Argon plasma cleaned glass coverslips were affixed to frame-seal slide chambers. They were then coated with poly-L-lysine solution and incubated for 30 min and then washed three times with filtered phosphate-buffered saline (PBS) buffer. Then aggregated Aβ42 present in sucrose fraction was added to the coverslips and incubated for 20 min before imaging using spectrally resolved PAINT[42]. The imaging was done using through objective (Plan Apochromat ×60 NA 1.49; Olympus) TIRF mode. A 532 nm solid state laser was used to excite the Nile red dye and the emitted fluorescence was passed through a dichroic mirror (Di02-R532-25 × 36) and a bandpass filter (F01-650/200-25). Nile red were dissolved into dimethyl sulfoxide to make a stock solution and then diluted using filtered PBS buffer to a final concentration of 1 nM for imaging. To obtain spectrally resolved images, a physical aperture (VA100/M; Thorlabs) which acts as slit and a transmission diffraction grating (300 grooves/mm; Thorlabs) were mounted just before the emCCD camera. Spectral calibration was performed using Tetra spec beads (Life Technologies).

**Separation of Aβ42 aggregates using sucrose gradients**. Aliquots of the aggregation reaction mixture of Aβ42 carried out in the absence of agitation were sampled at three different time points: (i) after 15 min (the end of the lag phase), (ii) after 30 min (the midpoint of the growth phase) and (iii) after 60 min (in the plateau phase). The samples collected from the different time points were mixed together and 400 μL aliquots were injected into a sucrose step gradient (10–50% with 10% increment w/v in SSPE buffer). The sucrose density in a given solution was determined by measurement of its refractive index. The sucrose solutions were centrifuged for 15 h at 24.696$g$ in an SW 60 Ti swinging-bucket rotor (Beckmann) at room temperature. The fractions were collected immediately and stored on ice until required.

**Atomic force microscopy imaging of single aggregates**. AFM studies were performed on freshly cleaved mica substrates. Ten microlitres aliquots of each diluted sample were deposited on the substrate at room temperature, incubated for 10 min, rinsed with 1 mL water, and then dried under a gentle flow of nitrogen gas. AFM maps were generated by means of a NX10 (Park Systems, South Korea) and JPK nanowizard2 system (JPK Instruments, Germany) operating in tapping mode and equipped with a silicon tip with a nominal radius of <10 nm. Image flattening and single-aggregate statistical analysis was performed by (Scanning Probe Image Processor) SPIP software (Image Metrology, Denmark). While the geometry of the AFM tip does not strongly affect the measurement of the height of an observed object, it is the primary determinant of the lateral resolution. Indeed, a tip with an apical radius of the same size, or larger than the dimensions of the object under investigation, will affect the measurement of the diameter because of the so-called convolution effect, resulting in an enlargement of the lateral dimensions. In the present experiments, objects with typical lateral dimensions within 0.3–10 nm were measured by a tip radius of approximately 10 nm. For this reason, we quantified the deconvoluted diameter of the aggregates[51]. The level of experimental noise (surface roughness plus electrical noise) for each image was measured using SPIP

software. It was well below 1 Å, leading to a signal to noise ratio ranging between 8 and 30. This allowed the cross-sectional height of individual objects within an AFM map as small as a monomer to be determined with high sensitivity and accuracy.

**Aggregation of dye-labelled Aβ42**. Monomeric Aβ42-labelled N-terminally with HiLyte™ Fluor 647 and HiLyte™ Fluor 488 were purchased from AnaSpec. Monomeric Aβ42 solutions were prepared by dissolving the labelled peptides in 10 mM NaOH at high concentration and then purified using a Bio-Sep 2000 HPLC column (Phenomenex) in SSPE buffer. Peak fractions were collected, and the peptide concentration was determined using the absorbance of the fluorescent label with an extinction coefficient of 250,000 $M^{-1} cm^{-1}$ for HiLyte™ Fluor 647 and 70,000 $M^{-1} cm^{-1}$ for HiLyte™ Fluor 488. Monomeric fractions were frozen immediately after size exclusion and kept at −80 °C until further use. For aggregation, aliquots of labelled monomeric Aβ42 were diluted to an equimolar concentration of 2 μM of HiLyte647 Aβ42 and HiLyte488 Aβ42 in low-binding Eppendorf tubes and incubated at 37 °C in SSPE buffer for aggregation under quiescent conditions.

**Single-molecule confocal FRET measurements**. Centrifuged samples were separated, and each aliquot were taken and diluted to nanomolar concentrations in buffer. Diluted protein samples were withdrawn through a single-channel microfluidic device (width = 100 μm, height = 25 μm, length = 1 cm), at a flow velocity of 50 μL/h to a syringe via polyethylene tubing. Single-molecule confocal FRET experiments were performed on the instrument described by Li et al.[52] by use of the 488 nm laser (1.5 mW) only. In brief, a 488 nm laser beam was directed to the back aperture of an inverted microscope (Nikon Eclipse TE2000-U) through a high numerical aperture oil immersion objective (Apochromat ×60, NA 1.40; Nikon). Fluorescence was directed onto a 50-μm pinhole, and then separated in two different channels by using a dichroic mirror (585DRLP; Omega Filters) and sent to two avalanche photodiodes (APD) (SPCM AQR-14; Perkin–Elmer Optoelectronics). For all of the single-molecule experiments, data were collected at 20 °C. A custom-programmed field-programmable gate array (Colexica) was used to count the signals from the APD and combine these into time bins. These were selected according to the expected residence time of molecules passing through the confocal probe volume. For each sample, data were collected for 10 min (100,000 time bins, bin width 0.2 ms). Confocal FRET data analysis was done using custom written software in IgorPro (wavemetrics)[53].

**Single-aggregate imaging with Thioflavin-T (ThT)**. To check if the aggregates of the Aβ42 peptide disaggregate in the presence of DesAbs, we have performed ThT based single-aggregate imaging using TIRF microscopy[41]. Glass coverslips were cleaned using an argon-based plasma cleaner and incubated with poly-(L)-lysine solution for 30 min for surface coating. Then Aβ42 aggregates only or Aβ42 aggregates and DesAbs were mixed with 10 μM ThT and added to the coverslips. The solution was incubated on the coverslip for 20 min to ensure attachment to the surface. For each sample nine different fields of view, which were selected in an unbiased way, were imaged. Each image was recorded at 100 ms exposure and 100 frames for each field of view. The number of ThT-active spots was calculated using ImageJ using constant noise tolerance for all the datasets.

**Design and production of the rationally designed antibodies**. Rationally designed antibodies (DesAbs) were generated using previously described procedure[15]. Briefly, complementary peptides were selected using the cascade method[14] to target linear epitopes within Aβ42 that scan its entire sequence. These complementary peptides were then grafted into the CDR3 of the DesAb scaffold by means of a mutagenic polymerase chain reaction with phosphorylated oligonucleotides (Supplementary Table 1). The DesAbs were then expressed and purified using the pRSET-B vector in Rosetta-gami 2 (DE3) (Merck Millipore). Cells were grown at 30 °C with shaking at 180 r.p.m. for 15 h using Overnight Express Instant TB Medium (Merck Millipore) supplemented with 100 μg/mL ampicillin. They were harvested, re-suspended in phosphate-buffered saline (PBS) supplemented with one tablet of EDTA-Free Complete Protease Inhibitor Cocktail (Roche) per 500 mL of cell growth, and lysed by sonication. Cell debris were removed by centrifugation at 15,000 r.p.m. (JA-20 rotor; Beckman Coulter) and the cleared supernatant was loaded on a $Ni^{2+}$-NTA Superflow column (Qiagen), previously equilibrated using PBS containing 10 mM imidazole. The column was then washed with PBS containing 40 mM imidazole. Finally, His-tagged DesAbs were eluted with PBS containing 200 mM imidazole. Imidazole was then removed using size exclusion chromatography with a HiLoad 16/600 Superdex 75 pg column (GE Healthcare). Protein concentration was determined by absorbance measurement at 280 nm using theoretical extinction coefficients calculated with ExPASy ProtParam. All the DesAbs used in this study are available from the author upon reasonable request.

**Cell cultures**. The BV2 cell line (RRID:CVCL_0182) was derived from immortalized murine neonatal microglia. They were grown in Dulbecco's modified Eagle's medium (DMEM) supplemented with 10% foetal bovine serum and 1% L-glutamine (Life Technologies) and incubated at 37 °C in a humidified atmosphere of 5%

$CO_2$ and 95% air, until a density was reached of approximately $1.6 \times 10^6$ cell/mL. All sucrose fractions were diluted 1:5 and added to the BV2 cells to perform the inflammatory assays. No significant cell death was observed over 24 h in any experiment.

**ELISA assays**. To determine cumulative TNF-α production, samples of supernatant were obtained after incubation of BC2 cells with early and late soluble aggregates of Aβ42 over 24 h and stored at −80 °C until analysed[28]. TNF-α was analysed using the Duoset® enzyme-linked immunosorbent assay (ELISA) development system (R&D Systems, Abingdon, Oxfordshire, UK). Three wells were used in each experiment to estimate error bars. RSLA was obtained from InvivoGen.

**Statistical tests**. To assess the statistical significance of the difference in the early and late aggregate-induced membrane permeability and inflammation, we performed two-sample *t*-test (unpaired) with three biological replicates (Fig. 1c, e) using origin 9.0. To determine if the inhibition of toxicity by the C-terminally binding antibody (DesAB 3–9) and N-terminally binding antibody (DesAb 36–42) are different, we also performed two-sample *t*-test (unpaired) at the highest antibody concentration using nine different fields of view (for membrane permeability assay) or three different wells (for the inflammation assay).

## Data availability

Data supporting the findings of this manuscript are available from the corresponding authors upon reasonable request. A reporting summary for this article is available as a Supplementary Information file. The source data underlying Figs. 1, 2, 4, 5, 6 and Supplementary Figs. 1, 2, 3, 4, 6, 7 are provided as a Source Data file.

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

## Acknowledgements

This study is supported by the Marie-Curie Individual Fellowship programme (to S.D.), EPSRC Studentship (to D.C.W.), Boehringer Ingelheim Fonds (to P.F.), Studienstiftung des deutschen Volkes (to P.F.), Senior Research Fellowship from the Alzheimer's Society, Grant Number 317, AS-SF-16-003, UK (to F.A.A), Swiss National Fondation for Science and Darwin College grant number P2ELP2_162116 and P300P2_171219 (to F.S.R.), Borysiewicz Biomedical Fellowship from the University of Cambridge (to P.S), the UK Biotechnology and Biochemical Sciences Research Council (to C.M.D.); the Wellcome Trust (to C.M.D), the Cambridge Centre for Misfolding Diseases (to P.F., F.A.A., P.S., C.M.D., and M.V.) and the European Research Council Grant Number 669237 (to D.K.) and the Royal Society (to D.K.).

## Author contribution

S.D., D.C.W., P.F., C.H., F.S.R., F.A.A., D.E., Z.X., D.R.W., J.A.V., P.S., and F.K. have carried out the experiments and analysed the data. T.P.J.K., C.M.D., C.B., M.V., and D.K. supervised the project. S.D., M.V., and D.K. conceived the idea, designed the study, and discussed the results. D.K. conceived the project. All authors discussed the results and wrote the paper.

## Additional information

**Competing interests:** The authors declare no competing interests.

