## [Peer Review File · Nature Communications]

This manuscript has been previously reviewed at another journal that is not operating a transparent peer review scheme. This document only contains reviewer comments and rebuttal letters for versions considered at Nature Communications. Mentions of the other journal have been redacted.

Reviewers' Comments:

Reviewer #1:

Remarks to the Author:

The authors have put together a nice response to the critiques and have included experimental attempts to address reviewer concerns. However, none of this data ended up in the manuscript, likely in part because it was not compelling and did not add to the study. The quality of AFM/TIRF images and the presence of potential artifacts has not been changed or improved. The bisANS binding should provide information on exposed hydrophobic surfaces but must be normalized for Abeta concentration or interpretation is not possible. Furthermore, bisANS should be compared between early/small/20% sucrose, late/larger/30% sucrose, and the 40% and 50% sucrose fractions. Recommendation is still major revision.

Reviewer #1

Comments: The authors have put together a nice response to the critiques and have included experimental attempts to address reviewer concerns. However, none of this data ended up in the manuscript, likely in part because it was not compelling and did not add to the study. The quality of AFM/TIRF images and the presence of potential artifacts has not been changed or improved.

Response: We would like to thank the Reviewer for the thorough review of our manuscript and for prompting us to include new data in the manuscript. We have added now new TIRF images in the main manuscript (4D and E) as well as in the supplementary information file which are as follows.

Supplementary Figure 5. Single aggregate imaging using Total internal reflection fluorescence imaging. This aggregated mixture of A β 42 collected at three different time points and was imaged (input) using TIRF in single aggregate level. The input was then loaded onto a sucrose step-gradient that is divided into fractions, and was subsequently imaged using TIRF microscopy.

The main aim of our AFM morphology mapping was the characterisation of the cross-sectional dimensions of the aggregates. In terms of resolution, it is important to realise that we are observing objects as small as individual monomers with size of approximately 4 kDa, which is at the limit of capabilities of AFM. To quantitatively demonstrate the quality of the acquired maps, we calculated the signal to noise ratio for each sample, defined as the average height of the aggregates in each map (signal) and the roughness of the surface (noise) (New Supplementary Figure 3). The average level of noise is well below 1 angstrom and the signal to noise ratio for measuring a single monomer is close to 10. Thus, the characterisation of cross-sectional height can be performed with high sensitivity and accuracy.

Supplementary Figure 3. Determination of signal to noise ratio of the AFM maps. Measurements enabled imaging of objects as small as individual Aβ42 monomers. We calculated the signal to noise ratio for each sample, defined as the average height of the aggregates in each map (signal) and the roughness of the surface (noise). The average level of noise is well below 1 angstrom and the signal to noise ratio for measuring a single monomer is close to 10 and reaches 30 for intermediate oligomeric species. Thus, the characterisation of cross-sectional height can be performed with high sensitivity and accuracy.

Comments: The bisANS binding should provide information on exposed hydrophobic surfaces but must be normalized for Abeta concentration or interpretation is not possible. Furthermore, bisANS should be compared between early/small/20% sucrose, late/larger/30% sucrose, and the 40% and 50% sucrose fractions. Recommendation is still major revision.

Summary of our results using bis-ANS fluorescence spectra of Aβ42 aggregates.

Response: We agree with the Reviewer that the results from the bis-ANS experiments do not provide a straightforward interpretation of the hydrophobicity of the individual aggregates present in sucrose solution until it is normalised by the Aβ42 aggregate concentration in each fraction. However, each fraction contains very small amounts of soluble aggregates which cannot be easily quantified using conventional ELISA assay, since they differ in structure and size, and therefore quantitative comparison among the samples cannot be performed.

To overcome this problem, we have used a newly developed assay from our group in which the hydrophobicity of individual aggregates can be determined. This experiment is concentration independent and therefore the hydrophobicity of individual aggregates can be compared between the different fractions. We therefore performed spectrally resolved super-resolution imaging - namely spectrally-resolved PAINT (points accumulation for imaging in nanoscale topography) - to map the surface hydrophobicity of soluble aggregates (*Bongiovanni et al. Nat. Comm. 2016, 7, 13544* and *Lee et. al. Nano Lett., 2018, 18, 7494*). We have used a solvatochromic dye, Nile Red, which transiently binds to protein aggregates allowing imaging over extended time. The emission spectra of Nile Red is sensitive to the local hydrophobicity resulting in a shift in the emission spectrum. If the spectrum is blue shifted, it indicates that the aggregates are more hydrophobic and vice versa. Therefore, this method can detect hydrophobic differences of protein aggregates in a complex mixture with different morphologies and measure the range of hydrophobicity of the different aggregates present as well as the average hydrophobicity. Using this technique, we also found that the aggregate present at 20% fractions are more hydrophobic than the aggregate present at higher sucrose fractions. We have added a section in the main manuscript and added the figure in supplementary section. (**Supplementary Fig. 5**). It was not possible to perform this experiment on the 50% fraction since there was a large amount of clumping of the aggregates in this fraction.

Supplementary Figure 6. sPAINT imaging of the surface hydrophobicity of the protein aggregates present in the different sucrose fractions. Frequency histogram of the sPAINT emission maxima from the individual protein aggregates present at 20%, 30% and 40% sucrose solution (total no of aggregate analysed 106, 145 and 67 for 20% 30% and 40% sucrose fraction respectively). Each fraction is diluted to 50 times before imaging using sPAINT. Aggregate clumping prevented measurements being made on the 50% fraction. Each distribution was fitted with a Gaussian function. The peak for 20% fraction shows a blue shift of approximately 10 nm compared to the 30% and 40% sucrose fractions.

Reviewers' Comments:

Reviewer #1:

Remarks to the Author:

The investigators have substantially strengthened an already intriguing study. The new data and analysis sufficiently helps to describe the aggregates formed at different times in the aggregation process and individually isolated in the sucrose gradient. I am fully satisfied with the additions to the manuscript and can recommend publication.